# The Coastal El Niño Event of 2017 in Ecuador and Peru: A Weather Radar Analysis

Rütger Rollenbeck [1,*], Johanna Orellana-Alvear [1,2], Jörg Bendix [1], Rodolfo Rodriguez [3], Franz Pucha-Cofrep [1,4], Mario Guallpa [5], Andreas Fries [4] and Rolando Célleri [2]

1 Laboratory for Climatology and Remote Sensing (LCRS), Faculty of Geography, University of Marburg, 35037 Marburg, Germany; johanna.orellana@ucuenca.edu.ec (J.O.-A.); bendix@staff.uni-marburg.de (J.B.); fapucha@utpl.edu.ec (F.P.-C.)
2 Departamento de Recursos Hídricos y Ciencias Ambientales, Universidad de Cuenca, Cuenca EC010207, Ecuador; rolando.celleri@ucuenca.edu.ec
3 Facultad de Ingeniería, Universidad de Piura, Piura 20009, Peru; rodolfo.rodriguez@udep.pe
4 Faculty of Engineering and Architecture, Department of Civil Engineering, Universidad Técnica Particular de Loja, San Cayetano Alto s/n, Loja 1101608, Ecuador; aefries@utpl.edu.ec
5 Subgerencia de Gestión Ambiental, Empresa Pública Municipal de Telecomunicaciones, Agua Potable y Alcantarillado de Cuenca (ETAPA EP), Cuenca EC010207, Ecuador; mguallpa@etapa.net.ec
* Correspondence: rollenbeck@lcrs.de; Tel.: +49-6421-282-5979

**Abstract:** The coastal regions of South Ecuador and Peru belong to the areas experiencing the strongest impact of the El Niño Southern Oscillation phenomenon. However, the impact and dynamic development of weather patterns during those events are not well understood, due to the sparse observational networks. In spite of neutral to cold conditions after the decaying 2015/16 El Niño in the central Pacific, the coastal region was hit by torrential rainfall in 2017 causing floods, erosion and landslides with many fatalities and significant damages to infrastructure. A new network of X-band weather radar systems in South Ecuador and North Peru allowed, for the first time, the spatio-temporally high-resolution monitoring of rainfall dynamics, also covering the 2017 event. Here, we compare this episode to the period 2014–2018 to point out the specific atmospheric process dynamics of this event. We found that isolated warming of the Niño 1 and 2 region sea surface temperature was the initial driver of the strong rainfall, but local weather patterns were modified by topography interacting with the synoptic situation. The high resolution radar data, for the first time, allowed to monitor previously unknown local spots of heavy rainfall during ENSO-related extreme events, associated with dynamic flow convergence initiated by low-level thermal breezes. Altogether, the coastal El Niño of 2017, at the same time, caused positive rainfall anomalies in the coastal plain and on the eastern slopes of the Andes, the latter normally associated only with La Niña events. Thus, the 2017 event must be attributed to the La Niña Modoki type.

**Keywords:** La Niña-Modoki; precipitation extremes; weather radar; precipitation climatology

## 1. Introduction

A common conception about tropical regions is the abundance of water supply. However, large parts of the tropical South American Pacific coast are characterized by aridity and a strong inter-annual variability of rainfall.

Ecuador is located in a transition zone between one of the wettest spots on earth and the coastal deserts of Peru. In Colombia, close to the northern border of Ecuador, several places are exceeding 10,000 mm of rainfall per year [1]. In North Peru, annual amounts go down to less than 20 mm [2]. In addition to this zonal gradient, a strong meridional rainfall gradient is present in South Ecuador and Peru: The entire Amazonian (eastern) parts of Ecuador and Peru are permanently humid all year long, while the south-western parts of Ecuador and the coast of Peru are fully arid.

ENSO (El Niño Southern Oscillation) is generally capable of overturning those average gradients and is a major driver for localized extreme precipitation anomalies, causing frequent floods, landslides and strong ecological, economic and social impacts [3,4]. Consequently, the dynamics and impact of the ENSO phenomenon was the matter of previous research in Ecuador and Peru [5–8] and many studies analyzed the associated large-scale atmospheric circulation changes [9,10]. Nonetheless, the 2017 expression of ENSO with its turn to a regional coastal El Niño in Ecuador and Peru in January 2017 came as a complete surprise to many of the involved institutions and large parts of the scientific community [11].

Starting in 2014, the buildup of warm water in the western Pacific pointed to a strong El Niño [12], and in the following years, several alerts were given, which finally culminated in an announced "Godzilla El Niño" for 2015/16 [13,14]. Indeed, the sea surface temperatures (SST) in December 2015 and January 2016 were considerably elevated (+5 to +6 K), although more in the central Pacific and not so much on the eastern Pacific coast [15,16]. In the Ecuador/Peru region, however, the actual strength of El Niño 2015/16 and the impact was much less than expected [17]. After the passing of that episode, a general cooling tendency towards weak La-Niña conditions was observed, leading to the expectation of below-average precipitation.

This expectation proved wrong with the onset of the first heavy rain events in January 2017 and the involved institutions and authorities quickly changed their assessment and warned about an upcoming episode of a coastal El Niño [18,19], reporting coastal SST anomalies of up to 4 K, while central Pacific temperatures remained close to average [20]. In January 2017, the SST at the northern coast of Peru (Paita, 05°04′S, 81°06′W) increased from 18 °C to 28 °C in just three weeks, reaching +5 K of anomaly, and remained at elevated values during February and March. Conditions favoring heavy to extreme rainfall events prevailed for three months, well in to April 2017, and the whole episode from January to April 2017 was categorized as one of the rare coastal El Niño events. Only a few of those have been observed in the last 100 years, and none were studied with the help of modern monitoring equipment like satellite observations, reanalysis tools and ground-based weather radar [14,21]. Due to the rapid shift from La Niña conditions in January, some sources used the term La Niña Modoki [22].

Several studies have analyzed the 2017 episode based on larger-scale datasets: Garreaud [14] reviewed the diagnostic value of the classical Southern Oscillation Index (SOI) and pointed out the role of the global wind-field anomalies above the Pacific, specifically the weakening of the Southeast-tradewinds. He concluded that the 2017 episode was most likely linked to a persistent anticyclonic anomaly over the southern Pacific inducing quasi-stationary atmospheric Rossby wave-trains termed as Pacific South American mode. Those wave trains can link pressure anomalies across the whole of the southern Pacific. Furthermore his analyses showed a historical (1948–2016) low in the strength of the SE-tradewinds and even episodes of northerly atmospheric flow.

Takahashi and Martinez [21] studied the historic 1925 coastal El Niño event, the strongest recorded before 2017 and number 3 on the scale of all El Niño events in the last 120 years. They found that downwelling oceanic Kelvin-waves occurring with "normal" El Niños are absent during coastal El Niño events. Furthermore, these authors point out the feedback between gradients of SST in the Pacific and the ITCZ (Intertropical Convergence Zone): They affect the strength and outreach of northerly winds in this region, which advect additional moisture from the Amazon and the humid northern parts of Ecuador. The extent of the 1925 event may be indicated by the annual total of almost 3000 mm, which fell in Guayaquil, Ecuador in that year, close to the amounts recorded in 1983 and 1998, the strongest El Niño impacts known so far. For 2017, Guayaquil, Ecuador reported around 1950 mm, which is almost twice the normal annual rainfall. In Piura, Peru, the very strong El Niño episode of 1982/83 brought 2273 mm (about 30 times the average) and 1640 mm (about 22 times the average) in 1997/98. Both episodes lasted about six months with strong impacts on population and the terrestrial and marine ecosystems. In 2014, 2015 and 2016,

the annual totals were 26.1, 72.8 and 157 mm respectively. However, during the coastal El Niño 2017, it reached 780 mm (almost 10 times the average) and the rainy season lasted about two and a half months.

The influx of Amazonian moisture during the 2017 episode is considered as a major source of water vapor for heavy rain events [23]. An oceanic rainband forms there, in combination with weakened tradewinds over the tropical eastern Pacific which contributes to the suppression of upwelling cold water, thus increasing SST.

Rodriguez-Morata et al. [24] published a detailed analysis of the extent of the 2017 episode, depicting the precipitation anomalies for all of Peru (but none from Ecuador). They rank the 2017 episode among the most extreme of the last 40 years, thus comparing it to the both well studied "super" El Niños of 1998 and 1983. An interesting observation in their study is the spatial progression of rainfall anomalies, which apparently first occur in the Andes and then gradually advance towards the coast. Finally, those anomalies move northwards, where they merge with the normal rain season and leave the dry Peruvian coastal plain in its normal arid state.

The interaction of westerly wind anomalies above the Pacific and increasing SST as a trigger for El Niño episodes is well known [25]. They also play a role along the coast, as was analyzed by [20]. They could show that, indeed, there was a wind-induced localized anomaly in sea level during February and March 2017, very likely contributing to the displacement of warm surface waters towards the north Peruvian coast, and by this, suppressing or limiting the normal upwelling of cold waters.

Vazquez-Patiño et al. [8] used complex networks to identify causal flows of matter and energy affecting the rainfall distribution in Ecuador. They found that the influx of atmospheric moisture in the lower troposphere (<1500 m a.s.l.) is a major factor for rainfall formation in coastal areas, while changes in the wind-field exhibit control on precipitation in the Andes. Additionally, they mentioned the role of vertical wind-shear on the seasonality of rainfall.

On larger scales, study [10] analyzed the main factors for extreme precipitation events along the Andes from 1°N to 20°S. For the equatorial Andes, they found a weak relation of ENSO modes with extreme precipitation, but intentionally left out the western Andean slopes, due to a lower data coverage there. They also stated that precipitation variability is strongly associated with the low-level humidity transport over the east Pacific and related moisture convergence and atmospheric stability variations. Furthermore, the importance of understanding the development of local convection and its relation with humidity transport from the Amazon was stressed.

A common limitation of all the mentioned studies is the lack of data linking the global scale to the local scale. They are based on climate reanalysis (ERA and NCEP; ECMWF-Re-Analysis and National Centers for Environmental Prediction), numerical modeling and point observations of the available monitoring networks from national weather services. Regarding the failure to predict the 2017 episode and the overestimation of the regional impact of the 2015/16 El Niño, Ref. [26] have identified a systematic bias of current coupled climate models in simulating eastern Pacific El Niño events due to problems with the simulation of SST anomalies and the Walker circulation. Some works have also integrated satellite data and [10] used a merged data product of satellite and ground observations. However, they also reported a lack of direct and integrative observation at the meso-scale in the equatorial Andes and the equatorial coastal plain. The high spatial heterogeneity of extreme rain events and the rapid and dynamic development of such storms cannot be recognized from this type of data.

In this study, we aim to fill this knowledge gap by using data from the first network of weather radars on the South American Pacific coast (RadarNetSur; [27]). The interaction of the complex topography, short and sudden changes of atmospheric flow patterns and the resulting localized peaks of extreme rainfall can only be detected with high resolution weather radar. Specifically, in Ecuador, there is a lack of deeper analyses of such events, although anomalous rainfall on the Ecuadorian coast is much more frequent than in

Peru [28] and very likely must be attributed to factors and drivers not always linked to the ENSO cycle [6].

For this study, new datasets were analyzed to obtain a more detailed climatology of the region, by combining high-resolution weather radar with traditional weather observations and comparing the results to current reanalysis data. The methodology reflects this goal by explaining the generation of derived products like anomalies, spatial statistics, atmospheric motion vectors and divergence fields. These data allow to shed light on local dynamics of rain events and their contributing factors. The Section 3 reports on the basic climatology of the five years of this study period. Then, the general spatial and temporal development of the 2017 episode is depicted, and selected events of large impact are analyzed in greater detail. Specifically, the development and progression of individual rainstorms has hardly been monitored in the tropical Pacific coastal areas, due to the lack of appropriate observational data. The new tool of ground-based remote sensing in this region by means of weather radar provides maps and statistics for the scientific understanding of local extreme events, atmospheric dynamics previously not available. Furthermore, it can support administrative planning and emergency response.

Finally, preconditions and dynamic factors during the 2017 episode will be discussed in the context of the large-scale analyses of other studies.

## 2. Materials and Methods

The Radar Network is a common initiative led by the University of Marburg, Marburg, Germany (Laboratory for climatology and remote sensing—LCRS) and includes the partners Universidad de Cuenca (UDC), Cuenca, Ecuador, the Empresa Pública Municipal de Telecomunicaciones, Agua Potable, Alcantarillado y Saneamiento de Cuenca, ETAPA EP (public utility company), the Universidad Técnica Particular de Loja (UTPL), Loja, Ecuador and the Universidad de Piura (UDEP) in Piura, Peru. RadarNetSur consists of three X-Band-scanning weather radars with a range of 60 to 100 km and a single elevation. Reflectivity data is recorded with a resolution of about 500 m on a 5-min time step. A detailed description of the technical background and the geographical setting is given in [29–31].

The study region is defined by the coverage of the three radar systems of RadarNetSur (RNS; Figure 1), the first operational radar network in South America covering the Pacific coast. The initial installation dates back to 2002, when a first X-Band radar (LAWR, Local Area Weather Radar) was used to monitor precipitation in the provinces of Loja and Zamora in Ecuador [6]. In 2014, this single observation was extended by a system in the western part of the Loja Province (Celica Radar, GUAXX; SELEX RS 120) and one year later by the Cuenca Radar (CAXX; same type), located in the Cajas National Park near Cuenca, Ecuador. In 2019, another SELEX system was installed in the Universidad de Piura/Peru and the name of the network was changed to RadarNetPlus. However, those data from 2019 are not yet included as they are out of the temporal scope for this study. Hence, the full radar coverage analyzed here reaches from 1.8°S to 4.9°S and from 80.5°W to 78.5°W (about 68,000 km$^2$; Figure 1). To avoid boundary effects, a slightly larger domain was chosen for the ancillary data reaching from 0°S to 7°S and from 82°W to 77°W. This larger domain is termed RNP-domain.

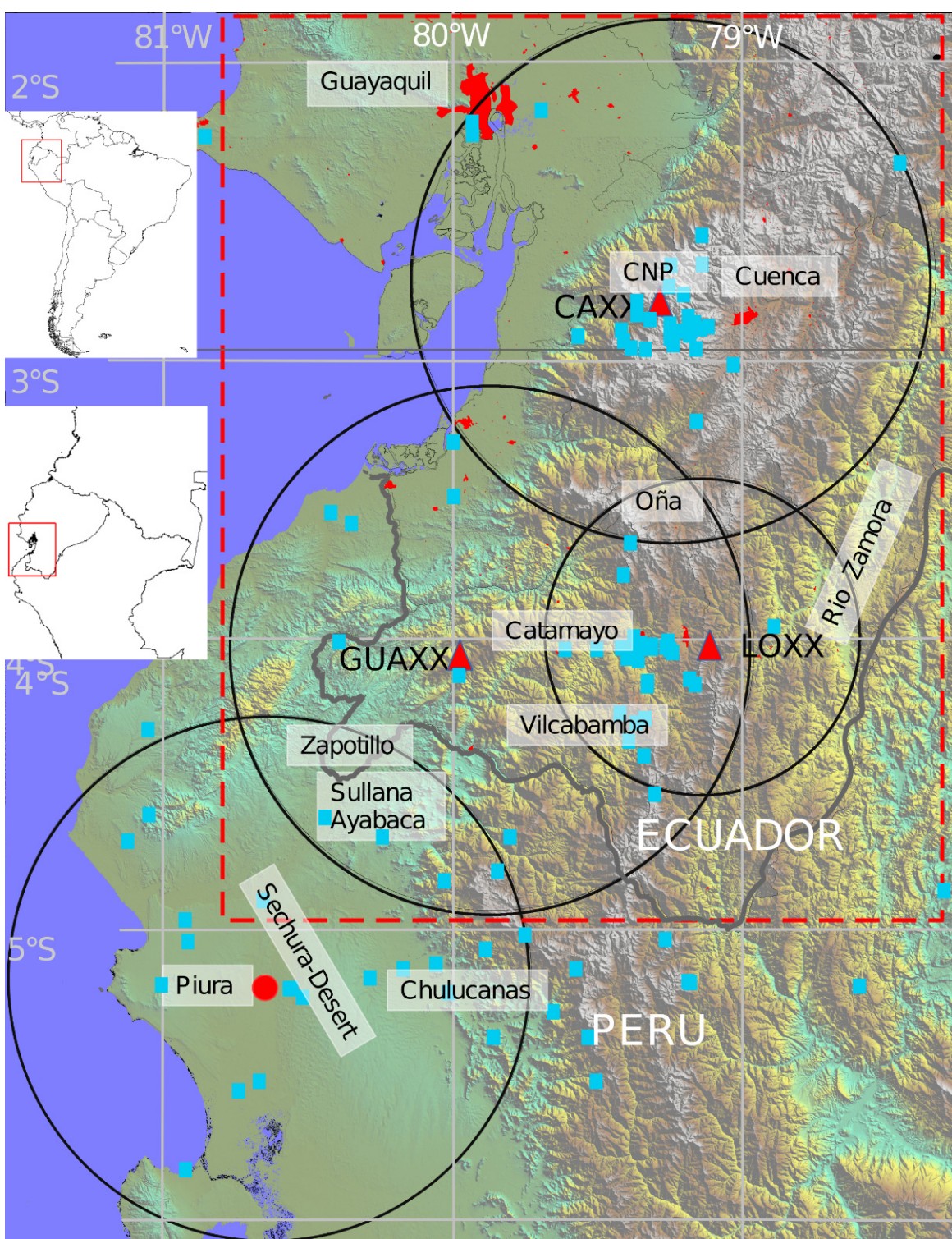

**Figure 1.** Map of the study domain. The red dashed frame is the RNS-domain for which radar data are available for this study. The black circles show the radar range and the red triangles are the radar sites used in this study. Highlighted location names (white frames) are referenced in the text (CNP = Cajas National Park near Cuenca, Ecuador). Red polygons show major cities; light blue squares are rain gauge sites used for calibration.

Additional data from several sources are integrated to enable the analyses of synoptic situations, oceanic conditions and relations to topography (Table 1). The time period 2014

till 2018 was selected according to data availability and to cover several years with strongly varying precipitation conditions.

**Table 1.** Overview of datasets used in this study.

| Institution | Data Type | Temporal Coverage | Spatial Coverage | Source |
|---|---|---|---|---|
| RadarNetSur | Radar precipitation data (Quantitative precipitation estimate QPE) | 2014–2018 | RNS-domain | RadarNetSur (RNS) |
| ECMWF | Reanalysis: temperature, specific humidity, latent heat flux, CAPE, wind-fields 850, 700, 500 hPa. 2014–2018 | 2014–2018 | RNP-domain | https://cds.climate.copernicus.eu (accessed on 7 October 2020) |
| GPM | Integrated Multi-satellite Retrievals (IMERG) | 2014–2018 | RNP-domain | https://gpm1.gesdisc.eosdis.nasa.gov (accessed on 15 July 2020) |
| LCRS, UDC, UTPL, UDEP | Hourly precipitation data | 1998–2018 | RNP-domain | RNS |
| INAMHI, SENAMHI, SYNOP | Daily precipitation data | 1998–2018 | RNP-domain | Mutual agreements/ http://www.ogimet.com (accessed on 9 January 2022) |
| UDEP | Hourly SST | 2012–2019 | Paita Port, Paita, Peru | RNS |
| SRTM | Digital elevation model | 2002 | RNP-domain | https://doi.org/10.5066/F7K072R7 (accessed on 2 April 2020) |

ECMWF: European Center for Medium range Weather Forecast. GPM: Global Precipitation Monitoring.

### 2.1. Radar Data Calibration

Raw reflectivity data from the three radar systems were submitted to a preprocessing scheme to correct for various radar-specific issues like noise and clutter contamination, beam blockage, atmospheric attenuation and hardware variations [29,30]. The corrected reflectivity data for each radar were then calibrated using 118 rain gauge sites as detailed in Table 1 by applying a space and time adapted parametrization of the Z—R-relation (Reflectivity vs. Rain rate).

Uncertainties for each radar system are addressed before compositing. Uncertainties range between 10–30% of daily rainfall at worst, if the rain gauges are considered as reference. The overall quality of the calibration gives an $r^2$ of around 0.85. Adjustment of raw radar-reflectivity is done individually for each system and handled separately from calibration. The aspect of attenuation is profoundly analyzed and addressed in several steps in the adjustment scheme. Aspects of radar drift are automatically compensated by using a daily variable set of calibration parameters.

All details of the calibration scheme are given in [29–31]. Finally, the calibrated QPE was combined into a composite image comprising all three radar coverages.

### 2.2. Precipitation Data Processing

Precipitation data coming from advanced weather stations were submitted to a quality control. Implausible values were flagged by error detection schemes based on variable thresholds, probability distributions, break point analyses and physical constraints as laid out in [32]. Flagged values were excluded from the dataset, no data reconstruction or spatio-temporal gap filling was employed. To build daily precipitation maps, all rain gauge data had to be converted to UTC as a common time reference. For some of the stations (mainly those from INAMHI and SENAMHI, which report daily at local time), this required the adaption of the temporal behavior of closely neighboring stations with higher temporal resolution. The maps were constructed in Python (Ver 3.8) by applying ordinary Kriging (PyKriging: http://pykriging.com; accessed on 10 June 2021) with a linear variogram and a nugget value of 0.1. Other variogram types were tested, but caused over- and underestimations of observed values [33]. Efforts to integrate additional drift terms like elevation or exposition have proven difficult and only applicable for rather small domains under more or less constant synoptic conditions [34].

Additionally, for some of the climatological interpretations, the whole period of 1998–2019 was considered, which is based on the interpolated rain gauge station data only.

### 2.3. Atmospheric Motion Vectors

A secondary use of the high-resolution radar imagery is the derivation of atmospheric motion vectors (AMV). For this purpose, objects or entities within the images have to be identified and their displacement for each time-step has to be traced. Recent progress in computer vision algorithms has made this a feasible task even for larger datasets, like the 551 × 701 pixel composite radar images produced at 5-min resolution in this study [35]. The OpenCV-library provides several methods to calculate such motion vectors. We tested different approaches all coming from the field of optical flow analysis like the Farnebäck, the Dense Inverse Search and the Dual-TVL1 method [36]. The latter turned out to be the most precise, which was proven by comparing predicted motions to actually observed motions and by minimizing residual image differences and residual vector differences. Moreover, it runs in a satisfying speed on a normal workstation. The Dual-TVL1 algorithm is based on total variation regularization and the robust L1 norm in the data fidelity term. By this, it offers increased robustness against brightness/reflectivity changes, occlusions and noise.

It has to be born in mind that atmospheric motion vectors represent the displacement (and spatial change) of rain fields and do not necessarily show the wind-field. Furthermore, the atmospheric motion is detected at varying altitudes, depending on the section of atmosphere illuminated by the radar beam. Last but not least, it can only detect motion, if there is sufficient rain to be observed by the radar. The initial elevation of the single beam of the three radars roughly corresponds to a pressure level of 700 hPa (Loja and Celica, Ecuador; beam elevation 0°), to 550 hPa (Cuenca; beam elevation −1°), but with increasing distance, it goes higher in the atmosphere, as earth curvature drops below the radar beam. On the other hand, these motion vectors are strongly indicative of internal dynamics of rain-generating clouds and, as such, very useful in understanding the development of rainstorms.

The ERA5 data products directly supply u- and v-vectors of upper atmospheric wind-fields, albeit at a much lower spatial and temporal resolution, and thus help to understand the synoptic circumstances under which rain events develop.

From the motion vector fields, spatial and temporal aggregates are produced for the whole domain and for parts of it, to clear up preconditions for heavy rain events. A further product used is the first spatial derivative of the motion vectors, which shows the divergence field of the rainstorms and, as such, can indicate the development stage of such events. This data product is especially useful for short-term case studies.

### 2.4. Synoptical Analyses of Extreme Events

To put the actual events of 2017 into context, firstly, a longer-term climatology was developed from the radar data between 2014 and 2018. The mean annual total for this period includes strongly varying years, like the relatively dry years of 2014 and 2016, but also two years with localized extremes (2015 and 2017), mainly in the coastal areas. Furthermore, the monthly mean for four equally divided sectors was calculated to derive a time series that shows the different regional behavior and impact of the interannual variations.

The central period of the 2017 episode of extreme rainfall was analyzed by deriving time series of daily maximum precipitation detected by the radars independent of the location of those values and identifying days exceeding the 5-year mean of highest daily rainfall by a value of 2sd (two times the standard deviation). This leads to the identification of the period from 21st of February till the 9th of March as being the most unusual rain period, of which five days were consequently submitted to case studies to unravel preconditions and formation of individual storm events.

In the case studies, the daily totals of QPE are compared to the synoptic situation, using the 850 hPa wind-field from ERA5, as well as the distribution of CAPE, surface

latent heat flux (slfh), potential temperature and specific humidity. Furthermore, the patterns of divergence are evaluated to identify dynamic processes within the storm cells in conjunction with the atmospheric motion vectors.

Relations between topographic factors, storm-cell motion, divergence and synoptic flow against the mean rain rate were derived for the week of strongest rainfall (2nd to 9th of March) in the area of strongest impact in the southwest of the radar domain to quantify the relative influence of the different factors. The results were finally summarized in a Pearson-correlation matrix for all factors.

## 3. Results

The high temporal and spatial resolution of the radar precipitation maps permits a much more detailed understanding of small-scale climatological features as well as the dynamic development of extreme events. To put the radar results into a larger context, additional analyses are supplied as Supplementary Material, partially referring to the larger RNP-domain covered by the ERA5 products and long-term monitoring (1998–2018) by means of the station network.

### 3.1. New High-Resolution 5-Year Baseline Dataset of Precipitation for South Ecuador and North Peru

As a background for the analysis of the 2017 episode, the baseline dataset is presented in Figure 2.

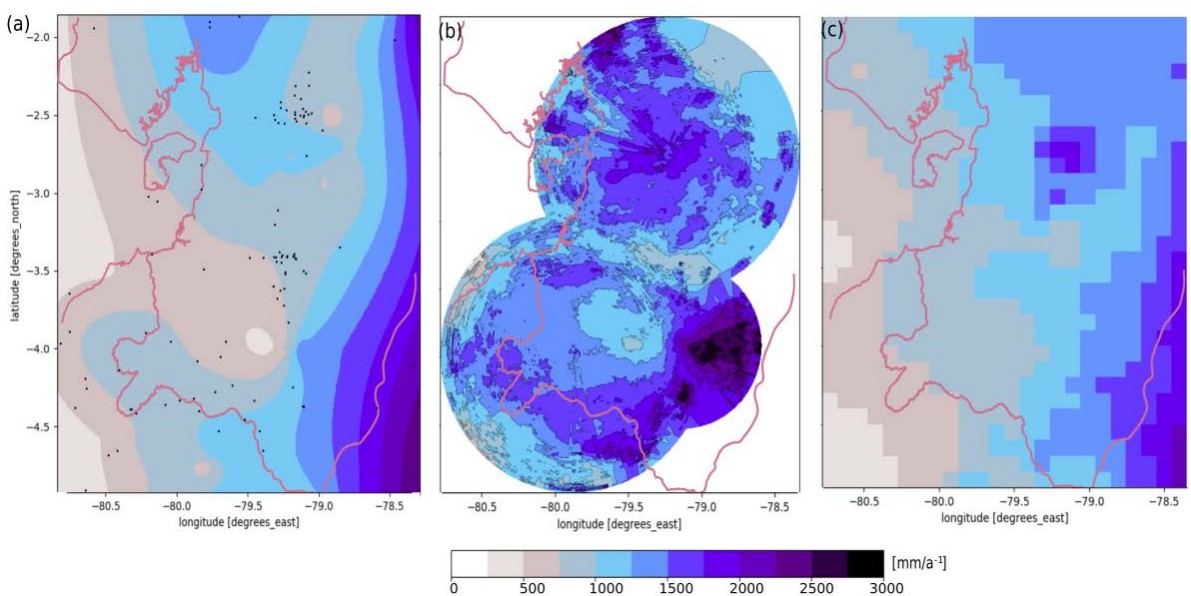

**Figure 2.** Comparison of the mean annual total (2014–2018) for the RNS-domain: (**a**) Interpolated (ordinary Kriging) station data from 118 rain gauge station (black dots); (**b**) Radar map; (**c**) GPM-IMERG, the radar map, shows areas of beam blockage to the NW of the CAXX-radar, which cannot be completely compensated at the annual aggregation level. The white lines in (**a**) delineate the sectors referred to in Figure 3.

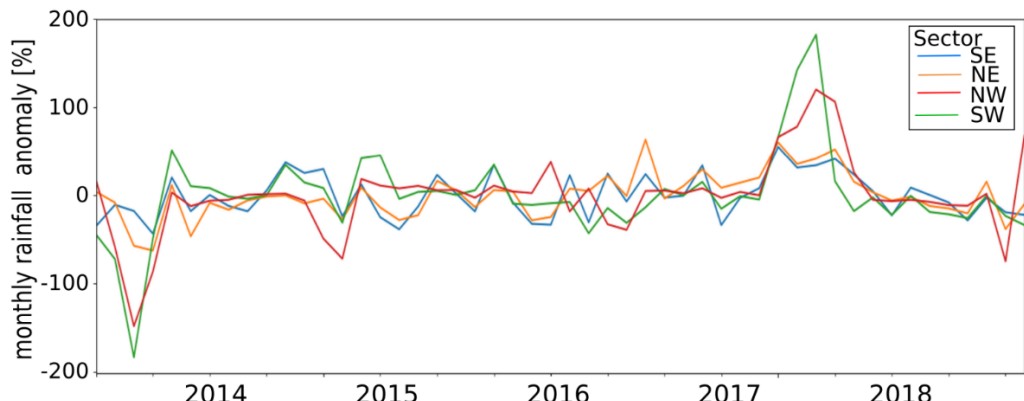

**Figure 3.** Relative anomaly of radar-derived monthly rainfall 2014–2018 (% of 5-year monthly mean) for the RNS-domain divided into the four equal sectors (Figure 2a).

Comparing the radar precipitation map of mean annual total precipitation to the interpolated station map (Figure 2), it is obvious that there is a strong underestimation of rainfall for many parts of the domain. While the interpolated map (Figure 2a) only shows few areas with totals up to 3000 mm, the radar map (Figure 2b) identifies many remote regions, especially in the higher mountains, where up to 3700 mm rainfall per year are observed. This is in line with several rain gauge observations which are not reflected in the interpolated map, due to the smoothing of the method. Additionally, the whole western part shows higher annual totals. The period considered here (2014–2018) included two heavy rain episodes, namely those of 2015 and 2017. Hence, they are probably not representative of the long-term climatological mean because the western half of the domain is normally drier. The GPM-IMERG data also shows higher values, especially in the high mountain areas, thus confirming that the gauge-interpolated product does not cover those areas in a sufficient manner. While both the interpolated map and the radar data correctly identify the dry valley of Catamayo, Loja, Ecuador. just below the center of the image, the GPM misses this feature completely. Similarly, the dry valley of Oña, situated between Cuenca and Loja, Ecuador at 3.5°S/79.5°W is only visible in the radar data and partially in the GPM-IMERG product. The interpolated map and GPM see higher rainfall in the high mountains of the Cajas National Park near Cuenca, Ecuador, but still underestimate the real amounts. Only the radar map identifies the deep influx of moisture from the east along the valley of Zamora and the orographic enhancement along the Ecuador/Peru border. Even the coastal mountains in North Peru and several mountain chains around Guayaquil show higher rainfall amounts appearing in Figure 2a,b, but not in Figure 2c. The radar map shows some sectors of strong beam blockage by mountains exceeding 5000 m towards the area of Guayaquil, Ecuador, which cannot be completely compensated for longer-term aggregations. Nevertheless, only radar data can reproduce all local features of the precipitation distribution, while the other products suffer from their low spatial resolution.

The eastern slopes and parts of the Andes mountains receive sufficient rain all year long, and in each of the observed years and the peak of the rainy season, rain shifts to May and June.

The south-eastern sector (Figure 3, blue line) largely comprising of the eastern slopes of the Andean mountains shows small variations from year to year with the most extreme month showing only a 50% deviation from the 5-year mean. To the north-east (orange line), variability shows stronger fluctuations, but still, all years are well below the 100% mark. The north-west sector (red line), mainly the province of Guayas in Ecuador and the western slopes of the Cajas National Park already exhibit the influence of the highly variable coastal climate, which then, in the south-west sector, (green line) reaches extremes of 200% negative and positive.

### 3.2. The 2017 JFMA Season

The spatially averaged time series of the four sectors of the RNS-domain in Figure 3 cannot really convey the full extent of the 2017 episode. Hence, in this section, anomalies of JFMA 2017 (the rainy season in the study region) in relation to the average conditions in JFMA (base period 2014–2018) are analyzed. Several spots in the arid landscape of the border region between Peru and Ecuador were hit by localized and much heavier extremes on shorter time scales. The highest anomalies were observed in the Ecuador/Peru border region and the coastal strip in North Peru up to the Gulf of Guayaquil. The interandean basins remained at normal levels of precipitation, while the slopes east of Cuenca as well as the eastern slopes of higher mountains in the south-east received additional rainfall up to about +100% of the 5-year mean. Negative anomalies were not observed.

No strict relation between terrain elevation and precipitation anomaly is visible in Figure 4, but the outreach of the coastal anomaly seems to be limited by the western slopes of the Andes mountains at about 80° W. What is different to canonical El Niño events is that normally the eastern Andes slopes experience drier conditions during such events and the positive anomalies occur in the coastal plain. In 2017, there were positive anomalies to both sides of the mountain chain, pointing to a mixture of larger-scale influences and regional processes. The Amazonian side of the mountains seems to experience the influence of the Pacific-wide cooler conditions with a slight positive precipitation anomaly, while the coastal (western) side shows the strong influence of the regional warming of the coastal waters.

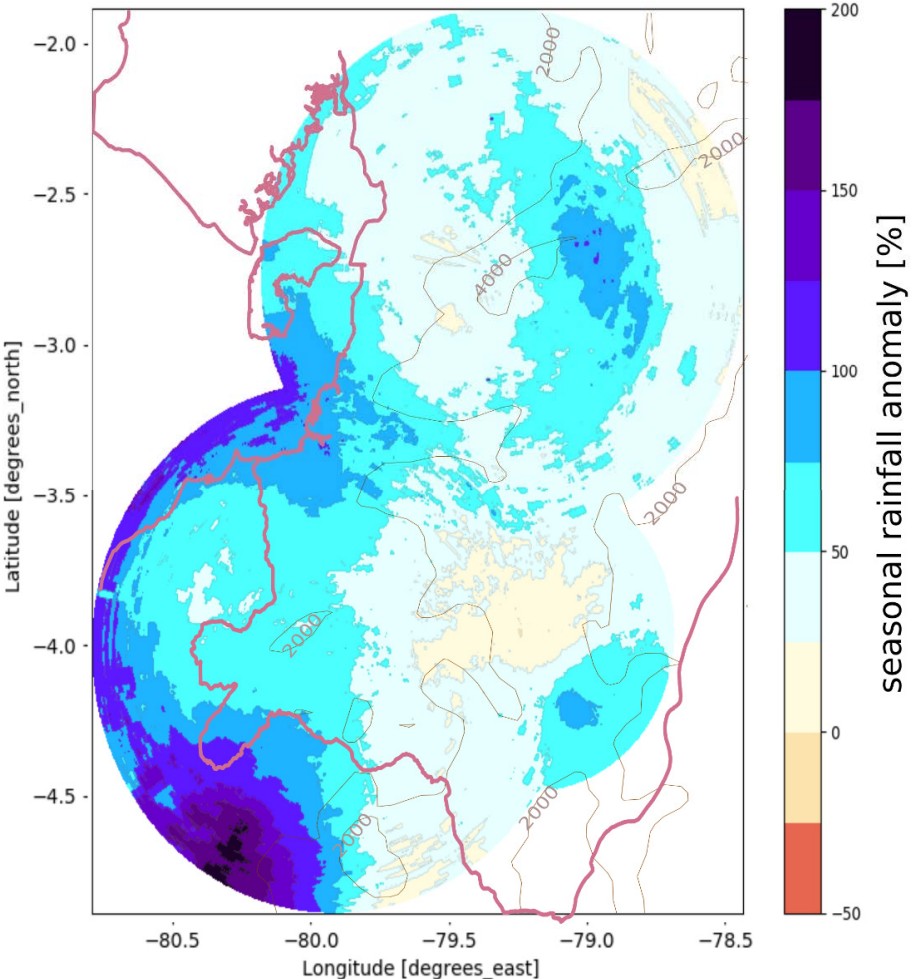

**Figure 4.** Anomaly map of radar precipitation of the 2017 JFMA season compared to the average behavior of the JFMA season from 2014–2018; 2000 and 4000 m terrain elevation contours (brown) are shown for topographical orientation.

### 3.3. Case Studies of Extreme Events

Although there is a general spatial pattern of the rainfall extremes, centered in the south-west of the study region (Figure 4), the dynamics of the 2017 coastal El Niño was shaped by a significant anomaly of SST in the Niño 1 + 2 region and exhibits a strongly episodic character as can be seen in Figure 5. Heavy rain events comprised of different weather situations with a variable spatial impact and distinguished spatio-temporal dynamics. They usually developed during the daily cycle, but some were persistent for several days, and in the worst case, always at a similar spot.

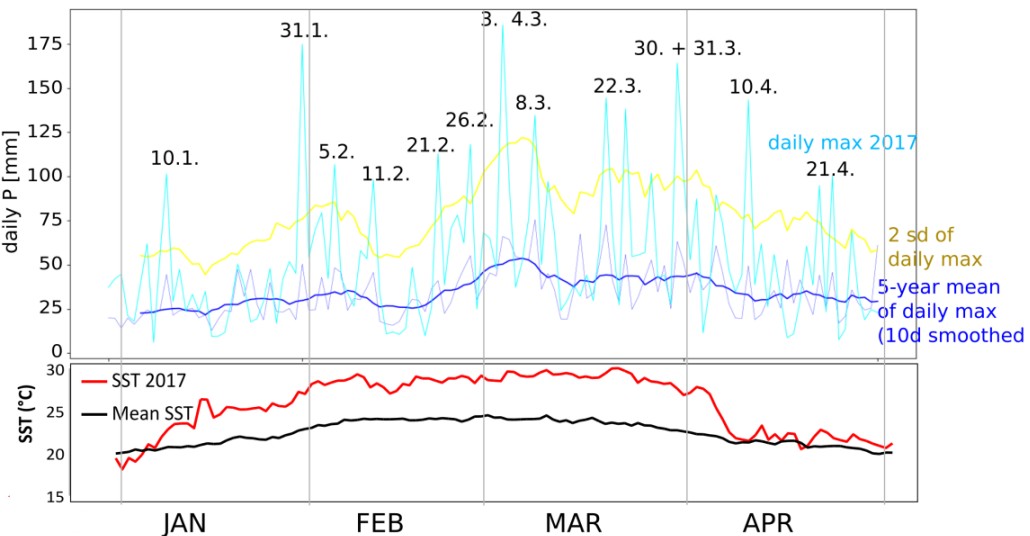

**Figure 5.** Temporal evolution of daily maximum rainfall (light blue) for the 2017 episode in the RNS-domain (above) and comparison to the base period 2014–2018: 5-year mean of daily maximum rainfall (thin dark blue line), 10-day smoothed daily maximum (bold blue line) and the range of 2 times the standard deviation (2sd) from the smoothed daily maximum (yellow line). Dates of peak rainfall are given as black numbers above the respective peak. Development of SST in the Niño 1 + 2 region measured at the port of Paita in Peru (below; base period for the mean is 1985–2014).

For places like the semi-desert provinces of Sullana and Ayabaca in Peru, this meant half of the normal annual precipitation fell in a single day (30 March 2017: 173 mm). It well exceeded any record of the very humid east Andean slopes and was only surpassed by three days during the Super El Niño 1998. The station Partidor in Peru reported rain for 28 days in March with a total of 1277 mm, where the long-term annual total is 650 mm. Chulucanas in north Peru. further to the south reached 1130 mm and many other stations exceeded the annual total in a few days of this month. In Piura, just south of the radar coverage, the dry weather of the first days of January 2017 changed to heavy rains in a few weeks (January total: 78.6 mm). The torrential rains continued during February (205 mm) and March (481 mm). The heaviest rainfall in Piura occurred on 22 March with 90.4 mm (more than the average annual total) causing an unusual increase of the Piura river discharge and reaching a peak value of 3468 m$^3$s$^{-1}$ on 27 March (Piura City, Peru), while the average discharge is below 100 m$^3$s$^{-1}$.

The first heavy events in 2017 occurred in the north of the study region in the Ecuadorian Guayas province. The events in the beginning of the rainy season (all dates in this section refer to the year 2017) mostly hit the northern and western part of the observed domain. On 10 January, the rain total of 101 mm struck the city of Guayaquil. On 5 February, again, Guayaquil received well above 100 mm of rain and 98 mm on the 11 February. After this period, the center of extreme rainfall shifted to the border region of Ecuador and Peru, mainly hitting the provinces of Sullana and Ayabaca in Peru and the county of Zapotillo in Ecuador. According to the radar data, eight times in five weeks, this region was struck by daily totals exceeding the 2sd-range of normal daily totals. The two days of 3 and 4 March

2017 brought 282 mm (196 and 86 mm) of rain, which is more than 50% of the normal annual total. To understand the short-term development, several events were analyzed in detail, of which three shall be presented as case studies in the following subsections. For these, the radar-derived variables of QPE, atmospheric motion vectors and divergence are shown in contrast to the synoptic wind-field at 850 hPa derived from ERA5.

### 3.3.1. Case Study 21 February 2017

On 21 February 2017, the first heavy impact in the lower mountain chain of the Sullana province, Peru occurred. The peak value of 113 mm daily total was not registered by any rain gauge but only visible in the radar data (Figure 6b). The rain event started on 20 February and in the next six days, several locations in that region accumulated rain amounts that were close to the annual total of around 300 mm. The general flow pattern visible from the ERA5-850 hPa reanalysis (Figure 6a) shows some interesting features embedded in a dominant westerly flow. While the coastal low-level jet still prevailed in the Sechura desert in Peru, its normal turn towards the east inverted and it impinged on the topography of the small mountains of the Sullana province. These mountains in Sullana and Ayabaca, Peru form a kind of convergent topography for such wind motions and thus induced an upward deflection. Additionally, there was a slight rotational movement within the wind-field. Looking at the atmospheric motion vectors from higher atmospheric levels and the divergence values, this rotational movement was associated with stronger convergence and thus contributed to the development of a stagnant dynamic low-pressure system. The radar data (Figure 6b) exhibit the typical appearance of a meso-scale cloud system with a diameter of about 80 km. On a broader scale, a strong influx of humid air masses from the north-east was visible in the atmospheric motion vectors. ERA5 data showed CAPE reaching 1200 J kg$^{-1}$ and specific humidity above 15 g kg$^{-1}$. The eastward flank of the storm cell showed divergent motion vectors (Figure 6c). Actually, the storm cell grew eastwards, against the prevailing easterlies in higher atmospheric levels.

This scenario set the stage for more than a full month of heavy rainfalls in this region. Almost each day from 21 February onward brought rain totals in the range of 40 to 100 mm to the Sullana region, Peru.

### 3.3.2. Case Study 2 to 4 March

The strongest event started on 2 March in the night hours. During the next day, a storm cell similar to that of 21 February developed at almost the same location, bringing 186 mm in 24 h and 96 mm on 4 March. Again, on 6 March, 50 mm was exceeded and 135 mm on the 8 March.

The actual development of the relevant atmospheric flows started before the onset of the rainstorm; hence the streamline plots are shown for 2 and 3 March. In the 850 hPa level, westerly flows dominated and the coastal low-level jet in north Peru showed a similar behavior as on 21 February (Figure 7). During the development of the storm, flow patterns turned eastwards towards the western Andean slope, and moist air from the warm coastal Pacific Ocean surface (SST anomaly +5.5 K, specific humidity > 15 gkg$^{-1}$) was advected to the region.

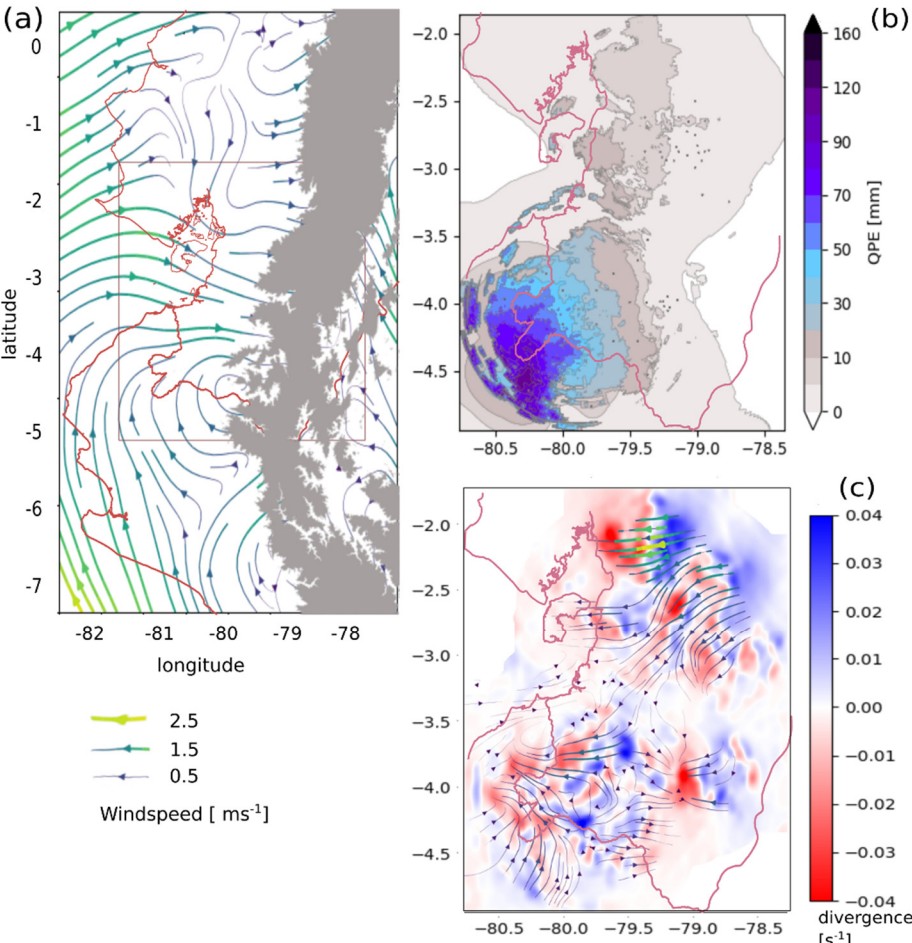

**Figure 6.** Event of 21 February 2017: (**a**) Lower tropospheric flow (ERA5-850 hPa wind-field); (**b**) Radar precipitation map and (**c**) Atmospheric motion vectors combined with divergence field. Note that red colors in (**c**) indicate convergent motion patterns.

The atmospheric motion vectors (Figure 8), however, show a different development at higher atmospheric levels: The general flow from the south on 2 March experienced a sudden shift to easterly flows during 3 March and this contributed to the formation of strong dynamic convergence in the most affected region.

These swift changes of atmospheric motion can be attributed to the growth of the one main storm cell, which brought the majority of rainfall as visible in the 15 min image time-series (Supplementary Material Figure S1). Starting on the 2 March at around 23:00 h local time, several smaller cells formed above the Sullana area, slowly progressing westwards where they converged with stagnant or eastward moving air. This convergent situation led to a rapid growth of the largest cell and it merged with the smaller storms to form a mesoscale cloud system (MCS) at 00:25 h local time. The cell brought peak rain rates well into 70 mm h$^{-1}$ and remained stagnant for more than 5 h before its intensity decreased with its slow progression towards the Pacific coast. The next MCS formed only 15 h later (Supplementary Material Figure S1), this time reaching further north and with slightly lower but still extreme rain rates.

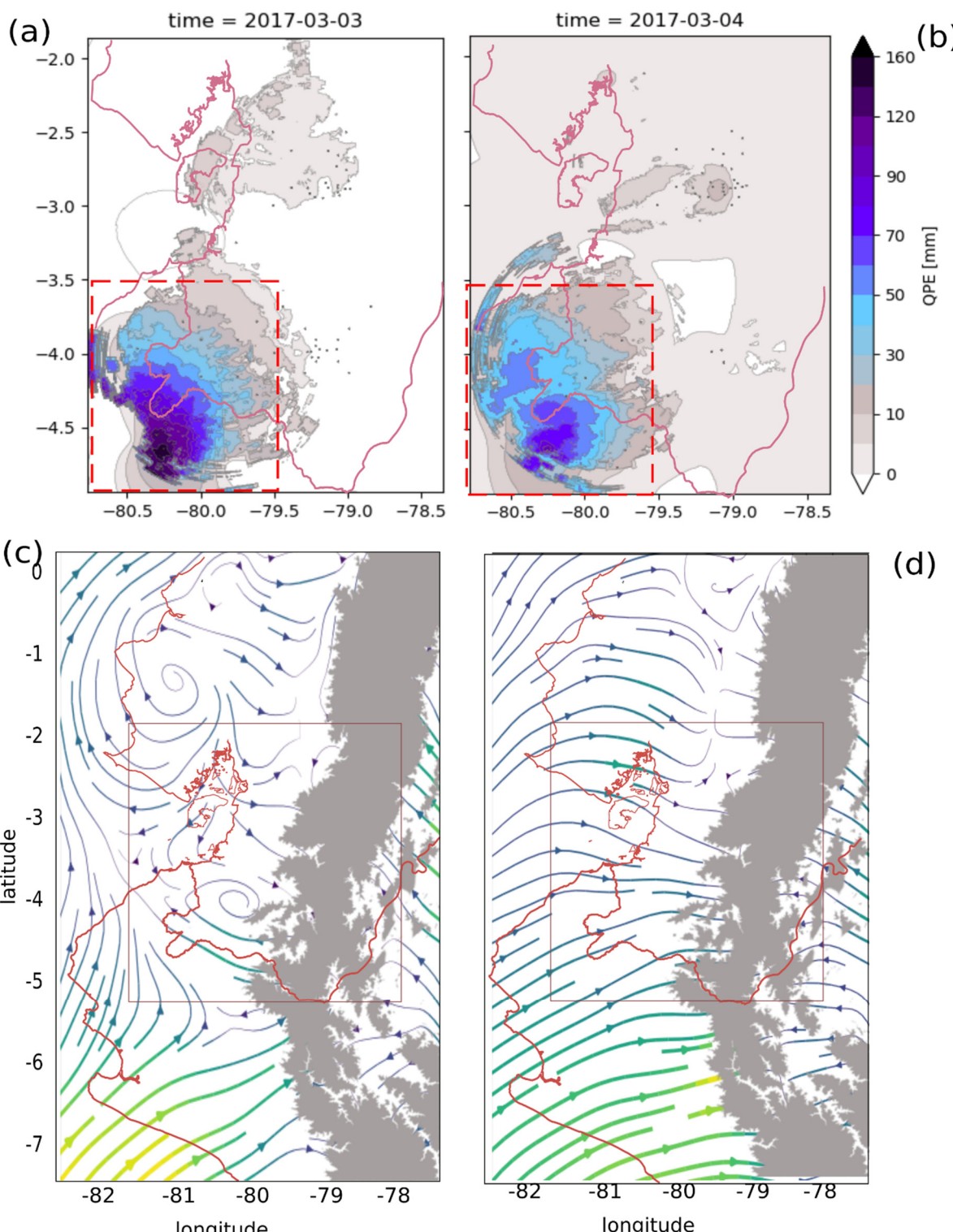

**Figure 7.** Radar precipitation maps for 3 (**a**) and 4 March 2017 (**b**). The red dashed box is the region analyzed in detail in Figures 11 and 12. ERA5-850 hPa streamline plots on 2 (**c**) and 3 March 2017 (**d**). Wind vectors use the same legend as in Figure 6.

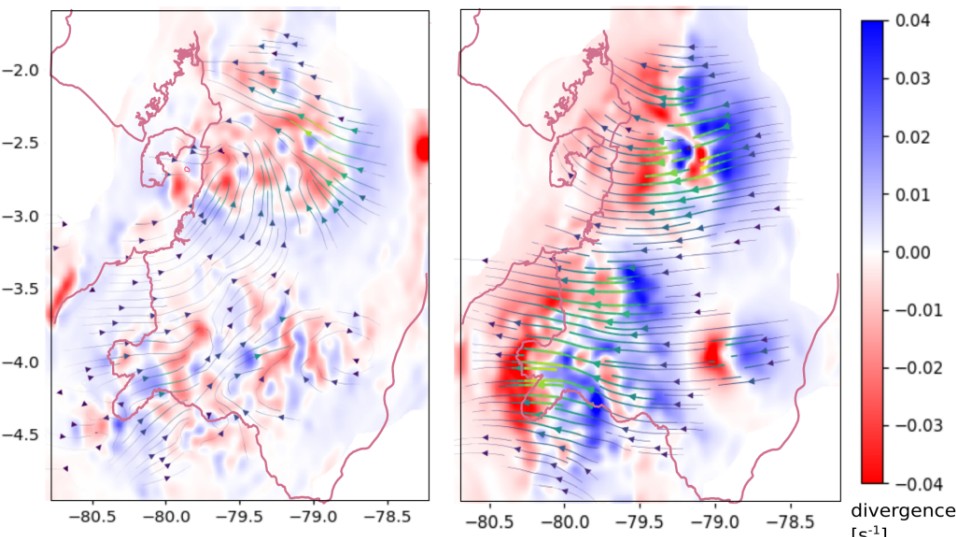

**Figure 8.** Atmospheric motion vectors and divergence fields on 2 (**left**) and 3 March 2017 (**right**) (RNS-domain). Wind vectors use the same legend as in Figure 6.

### 3.3.3. Case study 7 and 8 March 2017

On 7 and 8 March 2017, the extreme event started further north but quickly advanced towards the Sullana province again, where 125 mm in a single day was measured. This time, the whole coast was experiencing a strong westerly inflow in the 850 hPa level (Figure 9) and a line of strong rainstorms formed along almost 200 km from Guayas down to Peru (Figure 9a,b). The atmospheric motion at higher levels shows the advection of rapid easterly flow and strong convergence zones (Figure 10) just above the coast due to wind-shear blocking the outflow to the Pacific Ocean.

The easterly motion component of this storm development was even more rapid than on the previously depicted event and the convergence was even stronger. This can be seen in the image time-series (Supplementary Material Figure S2) at 22:05 local time (7 March), where the rapid westward progression of the trailing edge of the storm cell immediately stopped at about 80.5°W. The cloud system developed a bow-like structure thus causing the linear north-south extent of heavy rainfalls. It was followed by many smaller cells quickly advancing westwards in the strong wind-field of the higher atmosphere.

The further development of atmospheric flows continued with this opposing flow patterns of westerlies in the lower atmosphere and dominant easterlies at higher levels. Specifically, the common occurrence of lower atmospheric westerlies (850 and 500 hPa level (Supplementary Material Figure S3) seem to support the development of west wind bursts starting at the western Andes foothills and then progressing against the wind direction towards the coast in a few hours. This is indicative of coupled thermally-induced breeze systems (land-sea and mountain-valley breeze) as the initial impulse for convection.

In the second half of March, the strong influx of westerly air masses started to diminish with the reactivation of the southerlies of the coastal jet. Sea surface temperatures started to return to normal conditions (Figure 5). The strong convective activity began to recede towards the North and the last heavy events occurred on 10 and 21 April in the Guayas province, Ecuador, while in North Peru, the total for April was quite normal again with 6.5 mm in Piura, Peru.

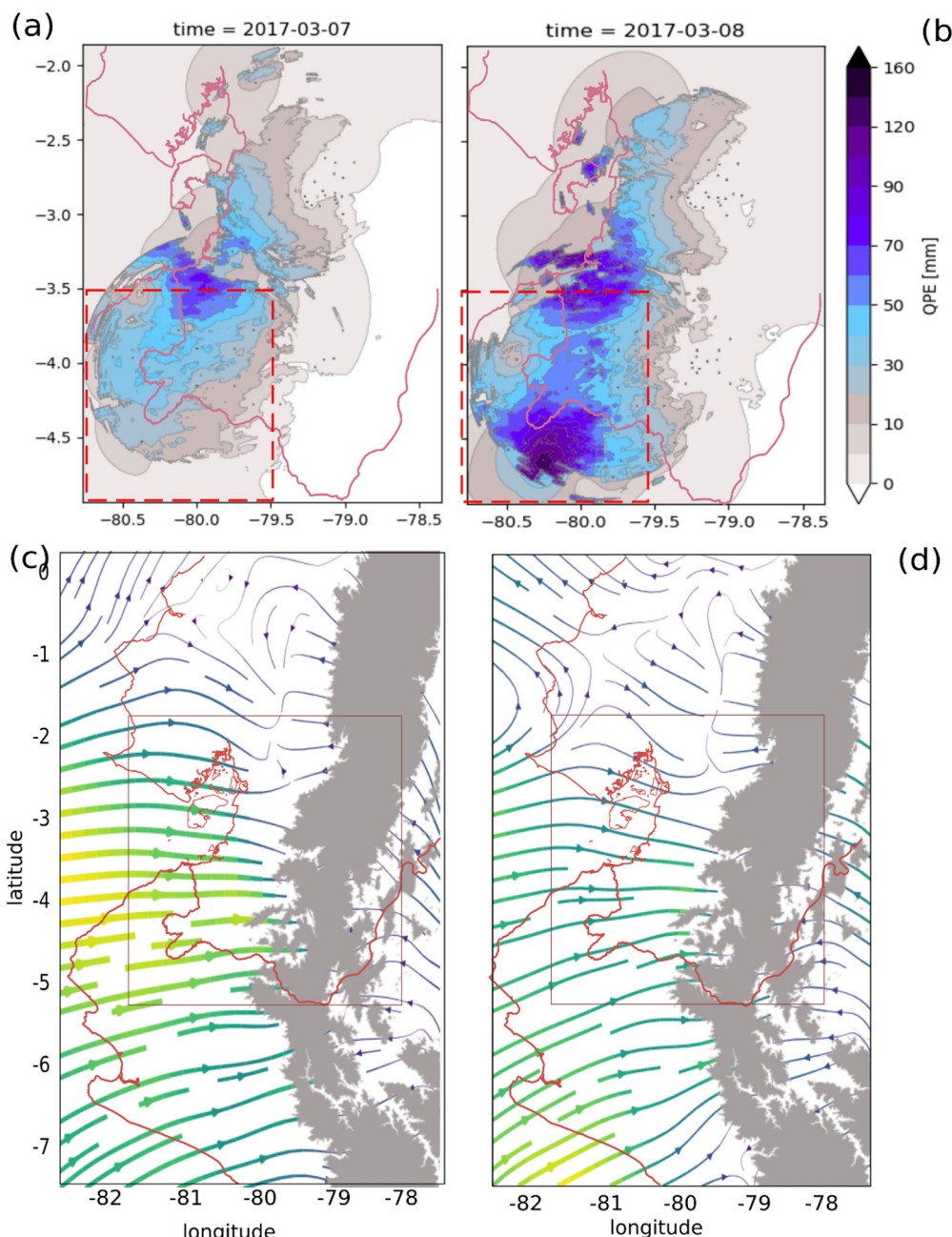

**Figure 9.** Radar precipitation maps for 7 (**a**) and 8 March 2017 (**b**) ERA5-850 hPa streamline plots on 7 (**c**) and 8 March 2017 (**d**). Wind vectors use the same legend as in Figure 6.

### 3.4. Relation of Extreme Precipitation to Topography and Synoptic Conditions

The case studies indicate some sort of characteristic development of rain storms in relation to topography, location and synoptic situation. The temporal development of the mean rain rate in the area of impact (dashed box in Figures 7 and 10) shows a typical diurnal cycle with the peak rain rate in the afternoon or evening (local time). In the morning hours, rainfall usually stops. Comparing terrain elevation to rain rate (Figure 11b), it is apparent that rainstorms initially occur at higher elevations and later extend towards the lower foothills to the west.

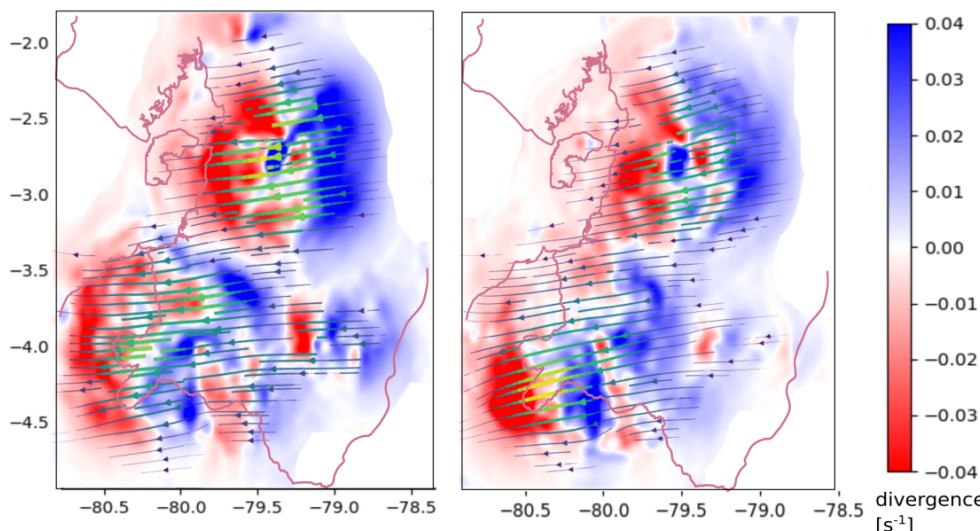

**Figure 10.** Atmospheric motion vectors and divergence fields 7 (**left**) and 8 March 2017 (**right**) (RNS-domain). Wind vectors use the same legend as in Figure 6.

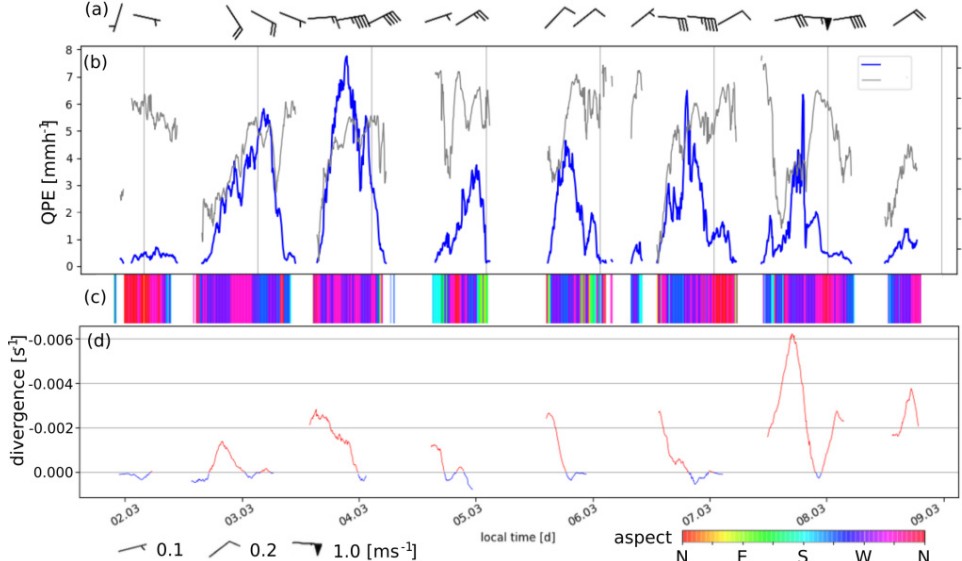

**Figure 11.** Time series of (**a**) spatial mean AMV velocity and direction (barbs); (**b**) Rain rate vs. terrain elevation (inverted scale) of raining pixels; (**c**) Aspect of raining pixels and (**d**) Divergence as spatial mean of the impact area shown in Figures 7 and 9. Wind barbs are scaled to ms$^{-1}$.

Similarly, rain events are largely centered above west-facing slopes (blue and red colors in Figure 11c), but in contrast, associated with a westward component in the 850 hPa field (not shown), as well as in the storm-cell motion vectors (Figure 11a). The displacement speed tends to increase in the decaying phase of the storms. In addition, rainfall starts with relatively strong convergence (negative values marked in red in Figure 11d) occurring at the mountain top but then decays rapidly during the rainstorm.

To summarize the relevance of the topographical factors and dynamic processes, a correlation matrix (Figure 12) shows the relative strength of the different influences.

The highest correlation is given for P vs. the easterlies (negative u_AMV) and aspect (north- and west-facing slopes). Divergence is negatively correlated to elevation, meaning that convergence occurs at mountain tops. Elevation as such, the v-component of AMV and 850 hPa wind, and the longitudinal (i.e., along the motion vectors) and vertical wind-shear exhibit no relation to rain rate during the analyzed week of extremes.

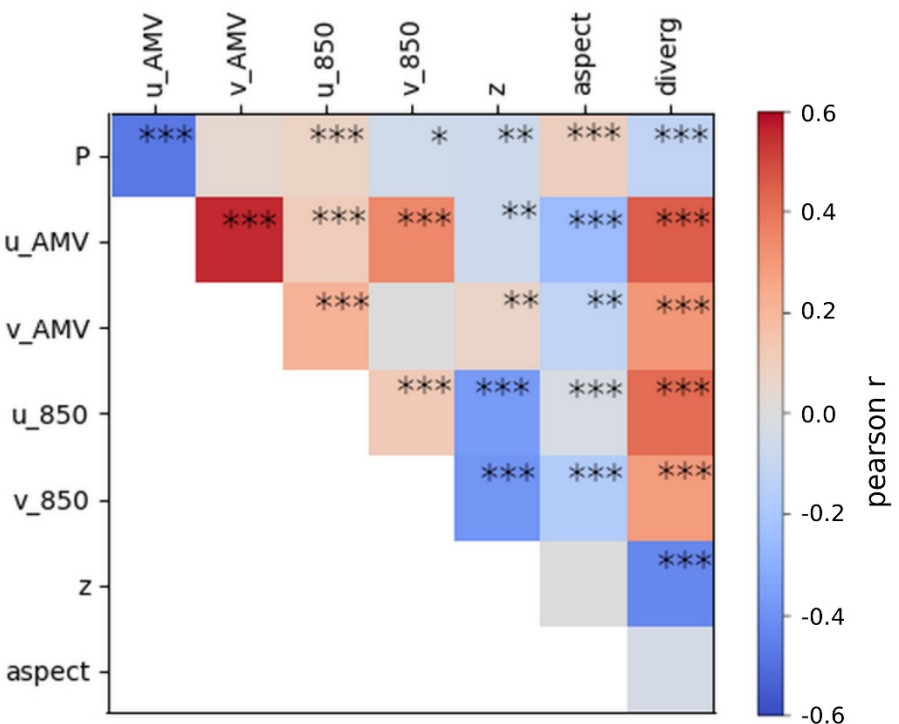

**Figure 12.** Correlation matrix (Pearson-r) of the relations between 5 min precipitation rate and analyzed factors and processes. Asterisks denote *p*-value (*** $p < 0.01$, ** $p < 0.05$, * $p < 0.1$).

## 4. Discussion

The analyses of five years of radar operation revealed important insight into local and regional patterns of rainfall distribution for the study domain. Spatial differences of interannual variability could be mapped in detail and process dynamics of extreme events unraveled. The lack of precipitation observations identified by [10] is well compensated by the new radar instruments of RadarNetPlus. The high amounts of rainfall in coastal mountain areas of South Ecuador and North Peru had not yet been reported, because they are caused by frequent smaller convective events being undetected by the relatively sparse observational station network. Currently used data products also fail to detect localized dry spots in the study region (e.g., the valleys of Catamayo and Oña) or features like the humidity transport along smaller depressions (valley of the Río Zamora, Ecuador). The high temporal resolution of the radar data also improves the observation of the development of episodic anomalies. The ENSO cycle is the main driver for such anomalies in Ecuador and Peru. El Niño episodes are well known to cause strong positive precipitation anomalies in the coastal plain and negative anomalies east of the Andes, while the interandean basins remain with more or less normal precipitation. La Niña episodes show the opposite behavior, with higher rainfall on the Amazonian side of the Andes and drier conditions at the coast.

Early in 2017 the global circulation was approaching ENSO neutral conditions, but the eastern Pacific behaved quite differently [3]. Normally, a strong coupling between rainfall at the Pacific coast and the Pacific ENSO indices is assumed. However, the major influence on the dry North of Peru and the arid parts of South Ecuador is related to the sea surface temperature (SST) in the smaller Niño 1 + 2 region [16,21,29]. This was certainly the case in 2017, but the eastern parts of the study domain showed the typical behavior of a normal La Niña episode, with increased rainfall on the Amazonian side of the Andes. So, the characteristics of the 2017 coastal El Niño exhibit a mixture of El Niño impact at the coast and La Niña impact east of the Andes.

At the beginning of the 2017 episode, the lower tropospheric wind-field above the coastal areas showed a southward shift of the westerly flows associated with the equatorial

counter current due to a weakening of the southern coastal low-level jet [4,14]. This weakening allowed the influx of additional moisture into the study region originating from the more humid coastal areas of North Ecuador. Additionally, it suppressed the normal offshore winds in the arid southern part of the study region. These offshore winds are a major mechanism of transporting the relatively high humidity of the lower atmosphere (on average 20–30 g/kg of q) towards the normally cold Pacific Ocean. This regional circulation mechanism normally inhibits the buildup of sufficient latent heat which could rupture the strong and almost permanent inversion layer above the coastal desert plain [2,37].

Similarly, study [38], in their statistical analyses of historical station data (1964–2010) point out the influence of the meridional wind components as a main factor for precipitation extremes for the western half of the Andes from 2°S down to 6°S.

With the further development of the 2017 episode, potential energy for convection accumulated above the coastal waters, as is visible from the ERA5 reanalysis data, but an additional trigger was required for actually inducing convection. Study [5] summarized potential mechanisms for rain extremes associated with these wind anomalies. During episodes with warm coastal waters, the diurnal cycle of the humidity-laden sea breeze forces air over the mountains of the western Andes slope, thus triggering local convection. This description is confirmed by our results, with the onset of such rainfall being late in the day and the initial impulse for west wind bursts starting at the west-facing slopes of the Andes. In agreement with personal observations, extreme events frequently started with orographically-enhanced convection in higher areas of the western ridges, leading to towering cumulus above the mountain chain. Higher cloud tops begin to travel westward, pushed by higher-level easterlies. This convective activity overcame the strong inversion layer by disturbing the descending motion on the leeward side of the Andes. With the progression towards the Pacific coast, additional humid air from the anomalous warm ocean was drawn in and storm activity extended to lower terrain. Although the initiation level of rain events varied in our case studies, peak rain rates were centered at the elevational zone between 700 and 1000 m a.s.l.

The development of the heaviest rain events was frequently associated with the formation of convergence zones due to opposing flow patterns between the mid (500 hPa) and lower tropospheric levels (850 hPa) [8]. As our case studies show, they occurred just above the higher terrain, where lower-level westerly winds encountered the continuous tropical easterlies and an intensive mixture of air masses happened. Together with the westward displacement of the storm cells, convergence zones were also displaced but lost their strength, so that rainstorm dissipated during the night. This further supports the assumption that thermal breezes are a relevant factor for triggering such events.

Regarding the regional perspective and in agreement with [24], a temporal transgression of heavy rainfall events from north to south could be observed during the episode of 2017. In our study, we could extend the perspective further north, with the observation of several heavy events in the Guayas Province, Ecuador at the start of the episode. Afterwards, the convective activity shifted to the south and remained centered for a long time in the provinces of Sullana/Ayabaca in Peru and the canton Zapotillo in Ecuador. Here, we conclude, that the topographical constellation with convergent ridges of smaller mountain chains add to the role of the shifting synoptic flow by guiding and concentrating humid air masses specifically to this location. The strongest period of the episode struck right here in the border region of Peru and Ecuador, a region that contributes main portions of water to the Piura and Catamayo-Chira river basin. As shown, this area received quantities in the range of two to three times the annual mean in just a few weeks. The resulting flooding then affected many areas downstream which are much more densely populated and under intensive agricultural use. Parts of Piura city, other towns and many agriculture areas of the Piura valley were flooded. While negative precipitation anomalies go widely unnoticed beyond the agricultural sector, the extreme rainfalls like those of 2017 cause much more of a societal impact, as they bring floods, landslides, inundations, infrastructure damage and, frequently, an increase of diseases and associated fatality rates [28].

The decay of the 2017 rain season began with a slow retreat of the heavy convective events towards the north and the last event was observed in the Guayas Province (21 April). There, heavy rainfall is part of the normal annual cycle and hydrology, which is why landscape and infrastructure are better adapted to the impact of high precipitation [28].

To summarize, three processes have to be considered as contributing factors for exceptional rain events: First, elevated SSTs in the Niño 1 + 2 region supply additional moisture in the lower troposphere of the coastal plain, thus raising the potential for moist convection [23]. Second, the interaction of the land-sea breeze with topographical factors like the mountain-valley breezes on the west Andean slopes induce air mass flow and thus forced lift and initial convection in higher elevation of the west Andean ridges [5]. Third, opposing atmospheric flows generate dynamic convergence zones and the mixture of humid air from the coast at lower levels plus the spill-over of Amazonian moisture causes the further buildup of storm cells sometimes growing into meso-scale cloud systems.

To better understand the special conditions which distinguish the 2017 coastal El Niño episode from past El Niño episodes like in 1997/98, a longer observational period is planned, possibly covering a canonical El Niño event. What can be concluded from the operational phase of RadarNetSur (2014–2018) is the fact that near-coastal conditions of SST and wind-field anomalies are more relevant for the regional development of precipitation anomalies than the warming of only the Niño 3, 4 region. Actually, regional coastal warming is much more frequent than Pacific-wide El Niño warming, as is evidenced by the 2008 episode [29], the 2015 episode [31] and now, the most recent 2017 episode.

Future work in RadarNetPlus will focus on deriving more general relations between the factors of SST, atmospheric flow and topographical constellation, also taking into account the normal course of precipitation formation beyond the extreme episode of 2017. With the integration of the Piura Radar data [32], the spatial extent can be further enlarged towards even drier and climatically more variable regions to the south.

**Supplementary Materials:** The following are available online at https://www.mdpi.com/article/10.3390/rs14040824/s1, Figure S1: Radar image time series 2 and 3 March 2017; Figure S2: Radar image time series 7 March 2017; Video S1: Radar image time series 1–4 of March 2017; Video S2: Radar image time series 6–8 of March 2017; Figure S3: 850 and 500 hPa zonal wind component 21 February–10 April 2017.

**Author Contributions:** Conceptualization, R.R. (Rütger Rollenbeck); methodology, R.R. (Rütger Rollenbeck) and A.F.; software, R.R. (Rütger Rollenbeck), A.F. and J.O.-A.; validation, R.R. (Rütger Rollenbeck); formal analysis, R.R. (Rütger Rollenbeck); investigation, all authors; resources, all authors; data curation, J.B.; writing—original draft preparation, R.R. (Rütger Rollenbeck); writing—review and editing, all authors; visualization, R.R. (Rütger Rollenbeck); supervision, R.R. (Rütger Rollenbeck); project administration, R.R. (Rütger Rollenbeck); funding acquisition, R.R. (Rütger Rollenbeck), R.R. (Rodolfo Rodriguez), J.B. and R.C. All authors have read and agreed to the published version of the manuscript.

**Funding:** Funded by Deutsche Forschungsgemeinschaft DFG under the grant number RO 3815/2-1 with contributions from Philipps-Universität Marburg, Marburg, Germany, Laboratory for Climatology and Remote Sensing (LCRS), Universidad Técnica Particular de Loja (UTPL), Loja, Ecuador, Empresa Pública Municipal de Telecomunicaciones, Agua Potable y Alcantarillado de Cuenca (ETAPA EP), Cuenca, Ecuador and Gobierno Provincial de Loja (GPL), Loja, Ecuador.

**Data Availability Statement:** Original research data will be provided at https://www.tropicalmountainforest.org and on request to the corresponding author, if not stated otherwise in the text.

**Acknowledgments:** The authors wish to thank the German research initiatives FOR402/FOR816, the technology transfer project RadarNetSur and RadarNetPlus (funded by Deutsche Forschungsgemeinschaft DFG and Philipps-Universität Marburg; DFG RO3815/2-1) for the implementation and operation of the radar network. We also acknowledge the contributions of the Gobierno Provincial de Loja and the Vice-rectorate for Research of the University of Cuenca (VIUC) through the project "High-Resolution Radar Analysis of Precipitation Extremes in Ecuador and North Peru and Implications of the Enso-Dynamics". The funders had no role in the design of the study; in the collection,

analyses, or interpretation of data; in the writing of the manuscript, or in the decision to publish the results.

**Conflicts of Interest:** The authors declare no conflict of interest.

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
