# Peer review of "The Coastal El Niño Event of 2017 in Ecuador and Peru: A Weather Radar Analysis"

_remotesensing, doi:10.3390/rs14040824_

Round 1
Reviewer 1 Report
The paper is very interesting, but there are things that will be improved. I suggest you that check the caption of all figures.
Now I list the different items that I found it:
- In line 57, the SST was elevated. How many degrees versus the averaged? In line 66 you indicate the variation, please indicate the variation here.
- Figure 1. Note that the raingauges are squares not dot, and you must indicate that the red triangle is the radar location. You could to increase zoom out in the subpanel to identify better the region.
- The table 1 need to be improved. The titles are cut and it's not easy to read.
- In point 2.1. are you doing programmed comparison from radar drift? In you calibration method, you have taken into account the attenuation of the rain on the radome. Remember that calibration and adjustment aren't the same. Please determine which process are you implanted. If you did calibration method, which is the uncertainty from each radar? This point is very important because you'll use this values as reference, especially if I look at the variations of the figure 2.
- In line 260, please include the units after the number. Before you commented that use three radars (LAWR,GUAXX and CAXX) but in this line only referees two locations... Which elevation angle is this first beam (PPI) in each radar?
- Figure 2, The units are mm/year? I don't understand the difference between a and b. There are zones that a indicates less of 500 mm/year and b indicates around 1500 mm/year, and I agree with you analysis that the smoothing method isn't good, but the station grid also isn't sufficient. This figure is the baseline of the dataset of 5 years, but which 5 years? From 2014 to 2018? Please include this information in the caption. If the period is 2014-2018, you comment that include 2 heavy rain episodes, so why do you use another period to obtain a good baseline?
- Figure 3. Please include title in y-axis. Why do you only show 4 years? The reference is "5-year monthly mean" from the RNS-domain. I suggest you details that RNS domain is the values obtained from the weather radar. In the figure shows different sector, that are defined in lines from 348 to 355. I suggest you creates a map where the reader shows clearly the sectors.
- Point 3.2. I think that JFMA is January, February, March and April, isn' it? This isn't season, because the seasons are winter, spring, summer and autumn where winter in astronomy field from 21/12 until 20 of march, and in meteorology field is from 1/12 until 28or29 of February. I suggest you change season to period, and define the period in the text.
- Figure 4. Mistake in caption "2000- and 4000 m-"! Include title in colorbar. I suppose that the reference is the same that the figure 3. Please indicate it.
- In line from 389 to 401 and figure 5. The text is'nt according with the figure 5. In the figure I isn't clearly that indicating the values on the peaks (light blue) and in any case there are 2 values, for example 3. and 4.3 very closer.... Please indicate the corresponding colour with the parameter in the caption, for example (light blue maximum daily precipitation, grey ... , yellow .. and blue ..). If the figure 5 shows the SW- sector, please detail it in the caption.
- In various cases March is written as Mar, please change this mistake.(examples lines 451, 453, 456, 467,....)
- In figure 7, please check the caption to solve the mistakes. There is a reference to figure 12, it is correct?
- In figure 8, please check the caption to solve the mistakes. The wind vectors are the same that Figure 6 or 9???
- In figure 10, the caption isn't complete.
- In figure 11, please indicate the corresponding panel to variable. for example a) is the AMV velocity and direction of wind, .....
Author Response
We thank the reviewer for his time and effort. For our detailed replies, please see the attachment.

Reviewer 2 Report
Thank you for providing me the opportunity to review the paper by Rollenbeck et al. on the precipitation event in 2017. I enjoyed the piece and I found the paper to be well organized and very well written. The key references are cited, the abstract is informative, the length is fine, the methods are appropriate and innovative, the graphics are publishable (more than publishable – they are very nice), and the results are interesting. I found the entire effort to be of the highest quality.
Author Response
We thank the reviewer for his effort and time and the encouraging words.

Reviewer 3 Report
A representative analysis that highlights the event La Niña Modoki from 2017, within ENSO that occured between 2014 and 2018, which marks new points, which are affected by heavy precipitations and, also, by a series of risk phenomena.
Author Response
We thank the reviewer for his effort and time and the encouraging words. Changes to the introduction are detailed in the attachment

Reviewer 4 Report
Manuscript ID: remotesensing-1571749
Title: The coastal El Niño‐Event of 2017 in Ecuador and Peru ‐ a weather radar analysis
Authors: Rütger Rollenbeck1*, Johanna Orellana‐Alvear1,2, Jörg Bendix1, Rodolfo Rodriguez3, Franz Pucha‐Cofrep1,4, Mario Guallpa5, Andreas Fries4, Rolando Celleri2
Summary: In this manuscript author tried to understand the coastal El Nino-Event of 2017 in the Ecuador and Peru region of South America using different ground-based, remote sensing, and reanalysis data sets. The manuscript is well organized, and the results are significant. Though the manuscript is clear most of the time, authors need to consider the below-mentioned minor comments before accepting the manuscript in the Remote Sensing journal.
Minor comments:
- Page 2, line 47: Replace “frequency causing floodings” with “ frequent floods”
- Page 2, lines 48-50: The sentence is unclear; please rephrase it.
- Page 2, line 55: “Where given” or “were given”?!
- Page 2, lines 69-74: Split the sentence into two.
- Page 2, lines 86-89: The sentence is unclear; please rewrite it.
- Page 2, lines 89-92: Split the sentence into two.
- Page 3, line 97: A space between the numeric value and the units is needed.
- Page 5: The black circle in Fig. 1 are not clearly visible.
- Page 5: In figure 1, I think “light blue dots” are “ light blue squares”
- Page 6, line 233: “Where tested” or “were tested”
- Page 7, line 285: What is “2sd”?
Author Response
We thank the reviewer for his effort and time. For a detailed reply, please see the attachment

Round 2
Reviewer 1 Report
Thanks for your answer. I understand that the paper is focused in climatological field, but I thought necessary give some details about the radar calibration, not only the different cites that you have used. Thanks for your effort.
Probably the format conversion has created some troubles, for this please check the next caption:
Figure 1 in my document appear dotssqaured??
Figure 5. The text in your answer isn’t the same that in my revised paper. Here is interesting that in your answer comments that in 4 of March there was 96 mm, but in the paper indicate 96 mm, please confirm the correct answer. Initially I didn’t understand that the values 3. And 4.3. are date, I encourage to indicate in the caption that the numerical values in upper panel are dates.
Figure 7 in my document continues appearing the 9….
The caption of Figure 10 still is missing.
In figure 11 the caption indicates the figures 7 and 10. I think that the corrects figures are 7 and 9. Could you checked? That appear in line 557. Please checked again.
In Table 1 the links doesn’t work. Please delete the hyperlink of correct it.
Author Response
Please see the attachment. And many thanks for the quick response!
